# Occurrence of *cfr*-Positive Linezolid-Susceptible *Staphylococcus aureus* and Non-*aureus* Staphylococcal Isolates from Pig Farms

**DOI:** 10.3390/antibiotics12020359

**Published:** 2023-02-09

**Authors:** Gi Yong Lee, Soo-Jin Yang

**Affiliations:** Department of Veterinary Microbiology, College of Veterinary Medicine and Research Institute for Veterinary Science, Seoul National University, Seoul 08826, Republic of Korea

**Keywords:** linezolid resistance, Q148K mutation in Cfr, *Staphylococcus aureus*, non-*aureus* staphylococci (NAS)

## Abstract

The emergence and spread of *cfr*-mediated resistance to linezolid in staphylococci have become a serious global concern. The acquisition of *cfr* confers multidrug resistance to phenicols, lincosamides, oxazolidinones, pleuromutilins, and streptogramin A (PhLOPS_A_ phenotype). However, occurrence of *cfr*-positive and linezolid-susceptible staphylococci has been identified. To investigate the mechanism underlying linezolid susceptibility in *cfr*-positive *Staphylococcus aureus* and non-*aureus* staphylococci (NAS) isolates from pig farms in Korea. Eleven *cfr*-positive and linezolid-susceptible staphylococci were analyzed for mutations in domain V of 23S rRNA, ribosomal proteins (L3, L4, and L22), *cfr* open reading frames (ORFs), and *cfr* promoter regions. The effect of the *cfr* mutation (Q148K) on the PhLOPS_A_ phenotype was determined using plasmid constructs expressing either the mutated (*cfr*_Q148K_) or nonmutated *cfr* genes. All 11 (six *S. aureus* and five NAS) *cfr*-positive and linezolid-susceptible isolates had a point mutation at position 442 in *cfr* ORFs (C to A) that resulted in the Q148K mutation. No mutations were detected in 23S rRNA, L3, L4, or L22. The Q148K mutation in Cfr is responsible for phenotypes susceptible to PhLOPS_A_ antimicrobial agents. To our knowledge, this is the first study to report the causal role of a single nucleotide mutation (Q148K) in *cfr* of *S. aureus* and NAS isolates in PhLOPS_A_ resistance. Continued nationwide surveillance is necessary to monitor the occurrence and dissemination of mutations in *cfr* that affect resistance phenotypes in staphylococci of human and animal origin.

## 1. Introduction

Linezolid, the first member of oxazolidinones approved solely for use in humans, has been considered a last resort antimicrobial agent in treatment of serious infections caused by antimicrobial-resistant Gram-positive pathogens, including methicillin-resistant *Staphylococcus aureus* (MRSA) and vancomycin-resistant enterococci (VRE) [1]. Because of its unique mode of action, which inhibits prokaryotic protein synthesis from a very early stage, and the synthetic nature of the drug, it was proposed that the development of resistance to linezolid is rare [2]. However, since the first report of linezolid resistance in a clinical isolate of MRSA in 2001 [1], the occurrence of linezolid-resistant *S. aureus* and non-*aureus* staphylococci (NAS) has been increased [3]. Moreover, although linezolid is strictly prohibited for use in livestock, the worldwide emergence of linezolid-resistant staphylococcal isolates has also been reported in various food-producing animals [4,5,6,7].

In staphylococci, linezolid resistance is mostly mediated by point mutations in the central loop of domain V region of the 23S rRNA [8,9]. As the primary target site of oxazolidinones, point mutations in these regions have previously been shown to be associated with the linezolid resistance phenotype [2]. In addition to the mutations in 23S rRNA, mutations in the genes encoding the 50S ribosomal proteins L3 (*rplC*), L4 (*rplD*), and L22 (*rplV*) at peptidyl transferase center (PTC) have been identified in linezolid-resistant bacteria [2,10].

Besides the point mutations in bacterial chromosomes, the transferable multidrug resistance gene *cfr*, encoding a 23S rRNA methyltransferase, has recently been reported [11,12]. The Cfr protein mediates methylation of the C8 carbon in the adenine residue at position 2503 (m^8^A2503, *Escherichia coli* numbering) in the 23S rRNA, resulting linezolid resistance [13]. Moreover, due to the proximal location of A2503 to the ribosomal binding sites of other antimicrobial agents targeting the bacterial 50S ribosome, the Cfr-mediated methylation in A2503 confers distinctive multidrug resistance (MDR) phenotypes to at least five classes of antimicrobial agents, including, phenicols, lincosamides, oxazolidinones, pleuromutilins, and streptogramin A (PhLOPS_A_) [14]. Since the first detection of *cfr* gene in a bovine isolate of *Staphylococcus sciuri* [11], carriage of *cfr* gene has been reported in various staphylococcal species in farm animals and farm environments [5,15,16]. In Korea, *cfr*-mediated linezolid resistance has also been identified in *S. aureus* and NAS isolates collected from pig farms and slaughterhouses [4,17,18].

Recently, it has been reported that staphylococcal isolates carrying the *cfr* gene failed to show resistance phenotype to linezolid [16,19,20,21,22,23]. In this study, *S. aureus* and NAS isolates that possess a *cfr* gene but are phenotypically susceptible to linezolid were identified in pigs and farm environments. Although the *cfr*-positive and linezolid-susceptible phenotypes have also been reported in *C. difficile* and *E. faecalis* [24,25,26], molecular mechanisms involved in the phenomenon are still unknown. Thus, to address the genotypic-phenotypic discrepancy in linezolid resistance in staphylococci, *cfr*-positive linezolid-susceptible *S. aureus* and NAS strains were analyzed for point mutations within the *cfr* open reading frame (ORF) and promoter sequences. Moreover, the impact of a point mutation in the *cfr* ORF on linezolid resistance was evaluated in a previously described plasmid system [27] expressing a point-mutated form of *cfr* gene derived from the *cfr*-positive linezolid-susceptible isolates.

## 2. Results

### 2.1. Identification of cfr-Positive Linezolid-Susceptible Staphylococci in Pig Farms

As shown in Table 1, a total of 11 staphylococcal isolates that were susceptible to linezolid but positive for *cfr* gene were obtained from the pig farms. Of the 11 *cfr*-positive linezolid-susceptible isolates, six were methicillin-susceptible *S. aureus* (MSSA) strains, all belonging to sequence type (ST) 398, and isolated from healthy pigs (SA16, SA17, SA18, and SA19) and farm environment (SA20 and SA21). The other five isolates were four different species of coagulase-negative staphylococci (CoNS), consisting of two *S. epidermidis* (SE9 and SE10) strains from pigs, two *S. sciuri* (SSC1 and SSC2) strains from farm environments, and one *S. simulans* (SSM1) from a pig. Except for the two methicillin-resistant *S. sciuri* strains, which had non-typeable SCC*mec*, all the other four CoNS were methicillin-susceptible isolates. MLST analyses revealed that the two *S. epidermidis* (SE9 and SE10) and one *S. sciuri* (SSC2) strains were ST570 and ST85, respectively. An MLST scheme has not yet been developed for *S. simulans*.

The six *cfr*-positive linezolid-susceptible ST398 MSSA strains displayed identical linezolid MIC of 2 mg/L, while *cfr*-positive linezolid-susceptible CoNS strains showed linezolid MIC range of 0.75 to 4 mg/L (Table 1). The SSM1 strain showed the lowest level of resistance to linezolid (MICs of 0.75 μg/mL) even with the carriage of *cfr* gene. All 11 *cfr*-positive linezolid-susceptible staphylococci were also carrying the *fexA* gene, which was correlated with the chloramphenicol resistance phenotype.

### 2.2. Genetic Assessment of cfr ORF and Its Promoter in LZD-Susceptible Staphylococci

Sequencing analyses of the *cfr* ORF revealed that all 11 *cfr*-positive linezolid-susceptible staphylococci had a single point mutation (C to A) at position 442 in the *cfr* gene versus the *cfr* sequences of linezolid-resistant staphylococci (SA2 and SE7 strains) (Appendix A). This point mutation results in a glutamine-to-lysine substitution at position 148 (*cfr*_Q148K_). In contrast, analyses of *cfr* promoter sequences (523-bp regions upstream of the ATG start codon) revealed no differences between the linezolid-resistant and –susceptible staphylococci. Moreover, no mutations in domain V of 23S rRNA or ribosomal proteins (L3, L4, and L22) were identified in the 11 *cfr*-positive linezolid-sensitive staphylococci (Appendix A). 

### 2.3. Impact of cfr_Q148K_ on PhLOPS_A_ Resistance Phenotype

As shown in Table 2, pRB474 constructs expressing either the point-mutated forms of *cfr* (*cfr_Q148_*_K_) or non-mutated *cfr* (wild-type *cfr*) were generated and then electroporated into the two linezolid-susceptible strains of RN4220 and ST398 MRSA SA100, which were negative for both *cfr* and *fexA* genes.

As expected, the *cfr*-positive SA12 strain showed resistant phenotype to linezolid (MIC of 16 mg/L) and all the other four classes of antimicrobial agents tested (Table 3). Complementation of the linezolid-susceptible *S. aureus* strains, RN4220 and ST398 MRSA SA100, with plasmids expressing the wild-type *cfr* resulted in 4-fold increase in linezolid MICs compared to those of the control strains carrying empty pRB474 plasmid (Table 3). In addition to the increase in linezolid MICs, expression of the wild-type *cfr* caused significant increases in phenicols, lincosamides, pleuromutilins, and streptogramins, leading to the PhLOPS_A_ resistance phenotypes. However, similar to the RN4220 and ST398 MRSA strains bearing empty pRB474 plasmids, the two strains complemented with plasmids expressing *cfr*_Q148K_ displayed susceptible or low-level resistance phenotypes to the five classes of antimicrobial agents.

### 2.4. Linezolid Population Analysis Profiles

For the two *S. aureus* strains (RN4220 and SA100) expressing the wild-type *cfr*, the linezolid population curves were significantly shifted to the right versus those for strains complemented with the empty plasmid or pRB4747::*cfr*_Q148K_ gene (Figure 1A,B). Area-under-the curve (AUC) values for linezolid population analyses were ~7-fold less for the two strains expressing *cfr*_Q148K_ gene than for the strains expressing the nonmutated *cfr* gene or carrying empty plasmids.

### 2.5. Genetic Environment of cfr_Q148K_


Four *cfr*-positive linezolid-susceptible staphylococcal strains (SA16, SA19, SE10, and SSC2) were selected for comparative analysis of the genetic regions harboring the *cfr*_Q148K_ genes. Genome sequence data indicated that the all *cfr*-positive strains possessed 38-kb plasmids (pSA16, pSA19, pSE10-1, and pSSC2-1) carrying *cfr*_Q148K_ genes. As shown in Figure 2, the identical structures of the *cfr*_Q148K_ containing regions flanked by the Tn558 transposon elements were found on the pSA16, pSA19, pSE10-1, and pSSC2-1 plasmids. BLASTn analysis showed that the 10-kb cfr-carrying segments of *S. aureus* (SA16 and SA19), *S. epidermidis* (SE10), and *S. sciuri* (SSC2) strains shared 99.9% nucleotides sequence identities. The *cfr*_Q148K_ harboring regions showed 99% nucleotide sequence identities with the previously reported wild-type *cfr*-carrying plasmid pSA12 (CP049977) of a linezolid-resistant ST398 MRSA strain SA12, except for sequence variations in transposon elements (*tnpA* and *tnpB*).

In addition, *fexA* genes flanked by the Tn*558* transposon elements were co-located in the downstream of *cfr*_Q148K_. Genetic analysis revealed that the *fexA* genes identified in this study were 100% identical in nucleotide sequence identity to those in the pSA12.

## 3. Discussion

Although linezolid resistance in staphylococci still remains rare, recent studies demonstrated a worldwide increase in the occurrence of linezolid-resistant *S. aureus* and NAS [4,7,17]. Unlike the chromosomal mutations in genes encoding 23S rRNA or ribosomal L3 and L4 proteins of linezolid-resistant isolates [9,34], the transmissible nature of *cfr*-mediated linezolid resistance in staphylococci has raised a significant concern in terms of horizontal transfer of resistance within and between different species of staphylococci [4,17]. In particular, carriage of the *cfr* on transferable plasmids and/or colocalization with insertion sequence (IS) elements has been reported in staphylococcal isolates of various animal and human origins [3,34]. Recently, the occurrence of *cfr*-mediated linezolid resistance in livestock-associated MRSA (LA-MRSA) and CoNS isolates obtained from pig farms and slaughterhouses was identified in Korea [4,17,18]. Whole genome sequence analyses of the linezolid-resistant LA-MRSA and CoNS isolates revealed that the *cfr* genes were located on plasmids and were usually associated with mobile genetic elements, such as transposons and conjugative elements [4,15,17,35]. 

In the current study, 11 *cfr*-positive but linezolid-susceptible staphylococci were identified in pig farms in Korea. As shown in Table 1, except for two methicillin-resistant *S. sciuri* isolates, 9 of the 11 *cfr*-positive linezolid-susceptible isolates were methicillin-susceptible staphylococci. Sequencing analyses of the *cfr* ORFs in these isolates revealed a specific point mutation of *cfr*_Q148K_. Although recently published studies have also described *cfr*-positive but linezolid-susceptible *S. aureus* [19,20,21,22,36] or NAS strains [16,23], the molecular genetic mechanism of the linezolid-susceptible phenotype in these strains has not yet been elucidated [2]. A recent study from Italy described a frameshift mutation within the chromosomal *cfr* gene (lack of adenine residue at position 379) in a *cfr*-positive linezolid-susceptible ST398 LA-MRSA of porcine origin [36]. Although we identified the *cfr*_Q148K_ mutation in previously published sequences of the MRSA strain SR153 [21] and *S. haemolyticus* strains VB5326 and VB19548 [34], the present study is the first to report the Q148K mutation in *cfr* ORF in association with a linezolid-susceptible phenotype in *cfr*-positive *S. aureus* and NAS isolates. Analyses of the *cfr* promoter regions in the 11 *cfr*-positive linezolid-susceptible staphylococci showed no variation in sequences compared to those of SA2, SA3, and SE7 isolates, suggesting that the linezolid-susceptible phenotype was not attributed to changes in transcription of *cfr* by *cis*-acting elements. 

Next, as shown in Table 3, the effect of the Q148K mutation in *cfr* ORF on susceptibility to PhLOPS_A_ antimicrobial agents was determined. The incorporation of wild-type *cfr* into the complementation plasmid, pRB474, resulted in increased MICs of all the five classes of PhLOPS_A_ antimicrobials in RN4220 and SA100 strains. In contrast, expression of *cfr*_Q148K_ from the same plasmid was unable to increase MICs to any of the PhLOPS_A_ antibiotics, indicating that the Q148K mutation in the *cfr* ORF is responsible for the linezolid-susceptible phenotype identified in the 11 *cfr*-positive linezolid-susceptible staphylococci. In addition, there were significant leftward shifts in linezolid population analysis AUCs in the *S. aureus* strains expressing Cfr_Q148K_ compared to the strains expressing wild-type Cfr (Figure 1A,B). Unlike the PhLOPS_A_ antibiotics, susceptibility to tetracycline and vancomycin was unaffected by the expression of wild-type *cfr* or *cfr*_Q148K,_ confirming the specific effect of *cfr* on PhLOPS_A_ agents. The impact of the Q148K mutation in *cfr* on PhLOPS_A_ phenotype is likely to be similar in NAS strains. However, attempts to electroporate the pRB474 constructs expressing wild-type *cfr* and *cfr*_Q148K_ into *S. epidermidis* or *S. sciuri* isolates were unsuccessful because of the much lower frequency of transformation in CoNS than in *S. aureus* strains [37]. 

WGS analyses of *cfr*-positive linezolid-susceptible *S. aureus* (SA16 and SA19) and NAS strains (*S. epidermidis* SE10, and *S. sciuri* SSC2) revealed the location of *cfr*_Q148K_ genes on 38-kb plasmids (pSA16, pSA19, pSE10-1, and pSSC2-1) in these strains (Figure 2), which showed >99% nucleotide sequence similarity. These results indicate that the plasmids possessing *cfr*_Q148K_ gene can be transmitted among staphylococci in the pig farm environments via horizontal transfer. Moreover, these plasmids showed >99% nucleotide sequence homology to the previously reported 38-kb plasmid, pSA12 [4], suggesting that the wild-type *cfr* and *cfr*_Q148K_ genes are located on the same plasmid backbone.

In line with the report by LaMarre et al., which reported low fitness cost of the *cfr* [38], there was no difference in growth rates between the *S. aureus* strains carrying the wild-type *cfr* and *cfr*_Q148K_ (Appendix A). 

Moreover, all 11 linezolid-susceptible staphylococci were positive for *fexA*. Previous studies from our laboratory and others reported that the *cfr* and *fexA* genes are frequently colocalized on a transferable plasmid [3,4,7,17,35]. In this study, WGS analyses also confirmed co-localization of *fexA* and *cfr*_Q148K_ on a transferable plasmid, indicating that *fexA* may contribute to the maintenance of the *cfr*_Q148K_-carrying plasmid under antibiotic selective pressure in pig farms. Although there was no significant difference in fitness costs between the wild-type *cfr* and *cfr*_Q148K_ without antibiotic selective pressure, future research is warranted to investigate effect of the mutation on long-term stability and transmissibility of the resistant plasmids, especially under selective pressure. Recently, the Q148K mutation within *cfr* ORF was detected in a linezolid-susceptible clinical MRSA isolate [21], suggesting that continuous monitoring on occurrence of *cfr*_Q148K_ and other *cfr* variants in staphylococci originated from human and animal is necessary. It should be recognized that our data in this study were generated from a limited number of staphylococcal isolates. Moreover, molecular mechanisms involved in resistance to other antibiotics were not included in this study. Nonetheless, this is the first to report the mechanisms underlying linezolid susceptibility in *cfr*-positive livestock-associated staphylococci isolated from pig farms in Korea. 

## 4. Materials and Methods

### 4.1. Bacterial Strains

Previous studies from our laboratory revealed the emergence of *cfr*-mediated linezolid resistance in livestock-associated MRSA (LA-MRSA) and NAS strains isolated from pig farms in Korea [4,17]. In a retrospective investigation of the prevalence of *cfr* among staphylococci on pig farms from 2017 to 2021, *cfr*-positive linezolid-susceptible *S. aureus* (n = 6) and NAS isolates (two *S. epidermidis*, two *S. sciuri*, and one *S. simulans*) were identified in a same pig farm. The staphylococcal isolates used in this study are listed in Table 1. Three linezolid-resistant *S. aureus* (SA2, SA3, and SA12) and one *S. epidermidis* (SE7) isolates were selected from the recently described staphylococcal strains isolated from pig farms [4,17].

All staphylococcal isolates were identified by using matrix-assisted laser desorption/deionization time-of-flight mass spectrometry (MALDI-TOF MS; Daltonics, Bremen, Germany) [39] and *tuf* gene sequencing (Bionics, Seoul, Korea) methods [40].

All *S. aureus* and NAS isolates were cultured in Mueller-Hinton broth (Difco Laboratories, Detroit, MI, USA) or tryptic soy broth (Difco Laboratories) for each assay. All isolates were stored at −75 °C until used for each experiment. 

For genotypic analyses of the *S. aureus*, *S. epidermidis*, and *S. sciuri* isolates, multilocus sequence typing (MLST) was performed as described previously [41,42,43]. The seven alleles of staphylococci were PCR-amplified, sequenced, and aligned to the MLST database (http://pubmlst.org/, accessed on 1 November 2022). For methicillin-resistant staphylococci, the presence of *mecA* and staphylococcal cassette chromosome *mec* (SCC*mec*) types were determined as previously described [44,45].

### 4.2. DNA Isolation and cfr Sequencing

PCR amplification of the *cfr* ORF and the *cfr* promoter region (523 bp upstream of ATG start codon) was carried out using the specific primer pair cfr-F (5′-GCGAAATGGCTCAATTTTCA-3′) and cfr-R (5′-TTCCACCCAGTAGTCCATTCA-3′) based on the sequence information of a *cfr*-harboring plasmid pSA12 (GenBank accession no. CP049977). DNA sequencing of the PCR product was performed at Bionics (Seoul, Korea). The presence of *fexA* gene, which encodes the florfenicol-chloramphenicol exporter, was determined in all isolates as described before [46].

### 4.3. Sequencing of 23S rRNA and Ribosomal Protein Genes

The presence of the previously described linezolid resistance-associated mutations in 23S rRNA or ribosomal proteins L3 (*rplC*), L4 (*rplD*) and L22 (*rplV*) [47] were determined by sequencing analyses using specific primer sets (Appendix A). *S. aureus* ATCC 25923 (GenBank accession CP009361), *S. epidermidis* ATCC 12228 (GenBank accession AE015929), *S. sciuri* NCTC 12103 (GenBank accession LS483305) and *S. simulans* NCTC 11046 (GenBank accession LS483313) were used as references for the primer design and sequencing analyses.

### 4.4. Genetic Manipulations and cfr Cloning

Genomic DNA samples from staphylococci were prepared as previously before [48]. Plasmid DNA was extracted from *Escherichia coli* and *S. aureus* using PureYield^TM^ plasmid miniprep kit (Promega, Madison, WI, USA). The preparation of competent cells and transformation of *E. coli* DH5α were performed as described before [49]. Electroporation of plasmid DNA into staphylococcal isolates was performed as described previously [50,51]. Briefly, 100 µL of electro-competent *S. aureus* cells were mixed with 1 µg of the plasmids, transferred into 1 mm electroporation cuvettes (Bio-Rad, Hercules, CA, USA), and kept on ice for 15 min. After application of electropulse at 2.3–2.5 kV, resistance 100 Ω, and capacity 25 µF, 1 mL of fresh TSB was added, and the cells were cultured at 37 °C for 1 h. The cells were then spread on tryptic soy agar (Difco Laboratories) plates containing chloramphenicol (10 µg/mL) and putative transformants were selected. 

For the *in trans* complementation constructs, nonmutated and point mutated *cfr* ORFs were PCR-amplified with primers cfr-HindIII (5′-CCCAAGCTTGCAAATTGTGAAAGGATGAAA-3′) and cfr-XbaI (5′-CCCTCTAGATCCACCCAGTAGTCCATTCA-3′) using purified DNA samples from SA12 and SA16 strains, respectively (Table 2). The PCR products were then cloned into the *Hind*III and *Xba*I sites of the pRB474 expression vector [30], which places expression of the cloned *crf* gene under the control of *veg*II promoter (Table 2). DNA sequences of the *cfr* ORFs ligated into pRB474 were confirmed by sequencing analyses and enzyme digestions.

### 4.5. Antimicrobial Susceptibility Testing

Antimicrobial susceptibility assays were performed according to the standard disc diffusion method described in the Clinical and Laboratory Standards Institute (CLSI) guidelines [31,32]. The antimicrobial agents used were ampicillin (10 μg), cefoxitin (30 μg), chloramphenicol (30 μg), ciprofloxacin (5 μg), clindamycin (2 μg), erythromycin (15 μg), gentamicin (10 μg), mupirocin (200 μg), quinupristin/dalfopristin (15 μg), rifampin (5 μg), trimethoprim-sulfamethoxazole (1.25–23.73 μg), and tetracycline (30 μg). The minimum inhibitory concentration (MIC) of florfenicol, chloramphenicol, clindamycin, linezolid, tetracycline, tiamulin, quinupristin/dalfopristin, and vancomycin was determined by the broth microdilution method or E-test^®^ (bioMérieux, Durham, NC, USA). Reference strains, *S. aureus* ATCC 25923 and ATCC 29213, were included for all disc diffusion and broth microdilution assays. The breakpoints for resistance to the antimicrobial agents were determined according to the CLSI documents, M100 [32], VET08 [31], and European Committee on Antimicrobial Susceptibility Testing (EUCAST) [33].

### 4.6. Population Analysis

Population analyses were carried out using linezolid as previously described with minor modifications [52]. Briefly, staphylococcal inoculum of ~10^8^ CFU/mL was plated onto Mueller-Hinton Agar (MHA) supplemented with different concentrations of linezolid (0.125, 0.25, 0.5, 1, 2, 4, 8, 16, and 32 µg/mL). The range of linezolid concentrations tested was selected to encompass sublethal to lethal linezolid levels based on the linezolid MIC data. After incubation at 37 °C for 24 h, visible colonies on the plates were counted. At least three independent assays were performed for each strain.

### 4.7. Whole Genome Sequencing Analysis

Whole genome sequence (WGS) data of *cfr*-positive linezolid-susceptible staphylococci were generated using a combination of Oxford Nanopore MinION (Oxford Nanopore Technologies, Oxford, UK) and Illumina iSeq (Illumina Inc., San Diego, CA, USA). Sequencing data were assembled de novo using Unicycler v.0.5.0. Functional annotation of assembled genome was carried out using the Prokka (v1.14.6) and Rapid Annotation in Subsystem Technology server tool. Integrative data from ResFinder (https://cge.cbs.dtu.dk/services/ResFinder/, accessed on 10 November 2022) of the Center for Genomic Epidemiology and the Comprehensive Antibiotic Resistance Database were used to confirm the presence and location of *cfr* genes. To analyze the *cfr*-containing regions, comparative sequence analyses were conducted on the strains sequenced in this study and the previously reported *cfr*-carrying plasmid pSA12 in a linezolid-resistant ST398 LA-MRSA strain SA12 (GenBank accession no. CP049977) [4]. 

The complete genome sequences of four *cfr*-positive staphylococci were deposited in the NCBI database: *S. aureus* SA16 strain (GenBank accession no. CP092999-CP093000) and SA19 strain (GenBank accession no. CP110318-CP110319); *S. epidermidis* SE10 strain (GenBank accession no. CP110320-CP110322); and *S. sciuri* SSC2 strain (GenBank accession no. CP093001-CP093009), respectively.

## 5. Conclusions

In the current study, we identified the occurrence of *cfr*-positive linezolid-susceptible *S. aureus* and NAS on pig farms in Korea. Our results suggest that (i) the Q148K mutation within the *cfr* ORF can recapitulate the linezolid-susceptible phenotype observed in the six ST398 MSSA and five CoNS strains (two *S. epidermidis*, two *S. sciuri*, and one *S. simulans*) collected from pigs and pig farm environments, (ii) the Q148K mutation in *cfr* confers susceptibilities to all the five classes of PhLOPS_A_ antimicrobial agents; and (iii) in addition to the *cfr*-mediated linezolid resistance, ST398 MSSA and NAS isolates likely acquire *cfr*_Q148K_-containing plasmids through intra- or inter-species interactions. 

## Figures and Tables

**Figure 1 antibiotics-12-00359-f001:**
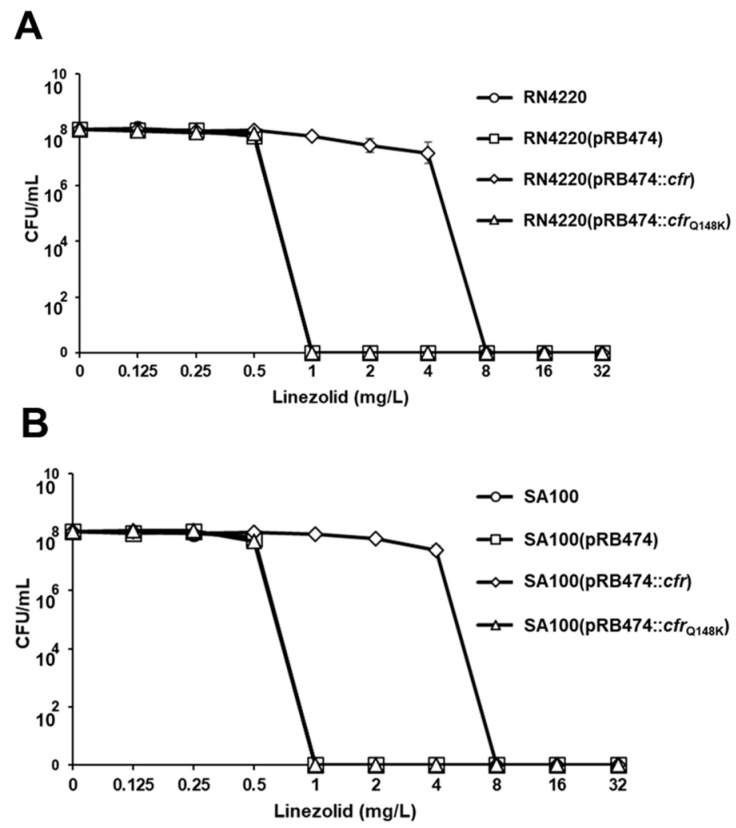
Linezolid population analyses in *cfr* transformants of RN4220 (**A**) and SA100 (**B**) strains. AUC values of RN4220 and SA100 strains expressing the wild-type *cfr* were 44.12 ± 1.48 and 45.69 ± 0.48, respectively. AUC values of RN4220 and SA100 strains expressing *cfr*_Q148K_ were 5.95 ± 0.03 and 5.90 ± 0.04, respectively.

**Figure 2 antibiotics-12-00359-f002:**
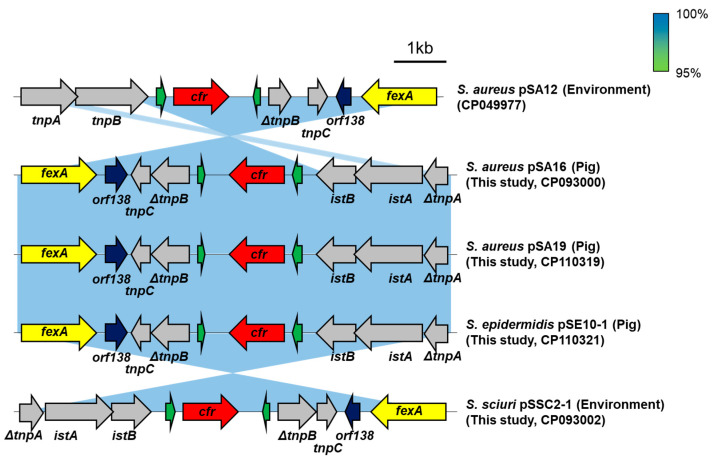
The schematic presentation of *cfr*_Q148K_ harboring regions in staphylococci strains. The *cfr*_Q148K_ genes are located on 38-kb plasmids (pSA16, pSA19, pSE10-1, and pSSC2-1) of staphylococci strains. Different colored arrows indicate that different genes with the direction of transcription; red arrows represent *cfr* genes; yellow arrows represent *fexA* genes; green arrows represent IS elements; blue arrows represent *orf138* genes; gray arrows represent transposon elements. Shaded regions indicate nucleotide sequence identities ranging from 95% to 100%.

**Table 1 antibiotics-12-00359-t001:** Genetic characteristics and antimicrobial resistant profiles of *cfr*-positive staphylococci isolated from pig farms.

Species	Strain ^1^	Origin	MLST-SCC*mec*	Methicillin Resistance ^2^	Antimicrobial Resistance ^3^	Positivity		MICs (mg/L) ^4^	Reference
						*cfr*	*fexA*	LZD	
CoPS *S. aureus*	SA2	Pig	ST398-V	MR	AMP-CEF-CHL-CIP-CLI-ERY-GEN-LZD-SYN-TET-TIA	+	+	12	[17]
	SA3	Pig	ST398-V	MR	AMP-CEF-CHL-CIP-CLI-ERY-GEN-LZD-SYN-TET-TIA	+	+	12	[4]
	SA12	Environ.	ST398-V	MR	AMP-CEF-CHL-CIP-CLI-ERY-GEN-LZD-SYN-TET-TIA	+	+	16	[4]
	SA16	Pig	ST398	MS	AMP-CHL-CIP-CLI-ERY-GEN-TET-TIA	+	+	2	In this study
	SA17	Pig	ST398	MS	AMP-CHL-CIP-CLI-ERY-GEN-TET-TIA	+	+	2	In this study
	SA18	Pig	ST398	MS	AMP-CHL-CIP-CLI-ERY-GEN-TET-TIA	+	+	2	In this study
	SA19	Pig	ST398	MS	AMP-CHL-CIP-CLI-ERY-GEN-TET-TIA	+	+	2	In this study
	SA20	Environ.	ST398	MS	AMP-CHL-CIP-CLI-ERY-GEN-TET-TIA	+	+	2	In this study
	SA21	Environ.	ST398	MS	AMP-CHL-CIP-CLI-ERY-GEN-TET-TIA	+	+	2	In this study
CoNS *S. epidermidis*	SE7	Pig	ST570	MS	AMP-CHL-CLI-ERY-GEN-LZD-SYN-TET-TIA	+	+	48	[4]
	SE9	Pig	ST570	MS	AMP-CHL-CLI-ERY-GEN-SYN-TIA	+	+	2	In this study
	SE10	Pig	ST570	MS	AMP-CHL-CLI-TIA	+	+	4	In this study
*S. sciuri*	SSC1	Environ.	NT-NT	MR	AMP-CEF-CHL-CLI-SYN-TET-TIA	+	+	2	In this study
	SSC2	Environ.	ST85-NT	MR	AMP-CEF-CHL-CLI-TET-TIA	+	+	4	In this study
*S. simulans*	SSM1	Pig	-	MS	CHL-CLI-GEN-TIA	+	+	0.75	In this study

^1^ SA2, SA3, SA12, and SE7 strains were *cfr*-positive linezolid-resistant staphylococci reported in previous studies. ^2^ MR, methicillin resistance; MS, methicillin susceptibility. ^3^ AMP, ampicillin; CEF, cefoxitin; CHL, chloramphenicol; CIP, ciprofloxacin; CLI, clindamycin; ERY, erythromycin; GEN, gentamicin; LZD, linezolid; SYN, quinupristin/dalfopristin; TET, tetracyclin; and TIA, tiamulin. ^4^ MIC, minimum inhibitory concentration. CoPS, coagulase-positive staphylococci; CoNS, coagulase-negative staphylococci; NT, non-typeable.

**Table 2 antibiotics-12-00359-t002:** Stains and plasmids used in this study.

Strain or Plasmid	Genotypic and Phenotypic Characteristics	Reference
*E. coli*		
DH5α	F-Φ80*lac*ZΔM15 Δ(*lac*ZYA-*arg*F) U169 *rec*A1 *end*A1 *hsd*R17(r_k_−, m_k_+) *phoA sup*E44 *thi*-1 *gyr*A96 *rel*A1 λ-; Host stain for transformation of plasmid constructs	[28]
*S. aureus*		
RN4220	8325-4, laboratory strain; accepts for foreign DNA	[29]
SA12	ST398-SCC*mec* V MRSA strain carrying *cfr*; LZD^r^	[4]
SA16	ST398 MSSA strain carrying *cfr*_Q148K_; LZD^s^	In this study
SA100	ST398-SCC*mec* V MRSA; *cfr* and *fexA*-negative, LZD^s^	In this study
Plasmid		
pRB474	*E. coli*-*S. aureus* shuttle vector; AMP^r^ and CHL^r^	[30]
pRB474::*cfr*	The wild-type *cfr* from SA12 cloned into pRB474	In this study
pRB474::*cfr*_Q148K_	Point-mutated *cfr* from SA16 cloned into pRB474	In this study

AMP^r^, ampicillin resistance; CHL^r^, chloramphenicol resistance; and LZD^s^, linezolid susceptibility.

**Table 3 antibiotics-12-00359-t003:** PhLOPS_A_ resistance phenotypes of *cfr*-carrying staphylococci and transformants.

Antimicrobial Agents (MIC)	MICs (mg/L) ^1^									
	SA12 ^2^	SA16 ^3^	RN4220	RN4220 (pRB474)	RN4220 (pRB474::*cfr*)	RN4220 (pRB474::*cfr*_Q148K_)	SA100	SA100 (pRB474)	SA100 (pRB474::*cfr*)	SA100 (pRB474::*cfr*_Q148K_)
Florfenicol (>8)	128	256	8	8	256	16	8	8	256	8
Chloramphenicol (≥32)	256	256	16	32	256	32	16	32	256	32
Clindamycin (≥4)	256	256	0.125	0.125	256	0.125	0.125	0.125	256	0.125
Linezolid (≥8)	16	2	2	2	8	2	2	2	8	2
Tiamulin (>2)	128	256	0.5	0.5	128	0.5	1	1	256	1
Quinupristin/dalfopristin (>4)	>32	2	0.38	0.38	2	0.38	0.38	0.38	2	0.5
Vancomycin (≥16)	1	1.5	1	1	1	1	1.5	1.5	1.5	1.5
Tetracycline (≥16)	>256	>256	0.25	0.25	0.25	0.25	>256	>256	>256	>256

^1^ MICs of *Staphylococcus* spp. indicate chloramphenicol, clindamycin, linezolid, quinupristin/dalfopristin, vancomycin, and tetracycline in CLSI [31,32], and florfenicol and tiamulin in EUCAST [33]. ^2,3^ SA12 and SA16 indicate staphylococci strains carrying *cfr* and *cfr*_Q148K_, respectively.

## Data Availability

No new data were created or analyzed in this study. Data sharing is not applicable to this article.

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
