# Peer review of "Occurrence of cfr-Positive Linezolid-Susceptible Staphylococcus aureus and Non-aureus Staphylococcal Isolates from Pig Farms"

_antibiotics, 2023, doi:10.3390/antibiotics12020359_

Round 1

Reviewer 1 Report

The authors thoroughly examine the mechanism underlying linezolid susceptibility in cfr-positive Staphylococcus aureus and non-aureus staphylococci (NAS) isolates from pig farms in Korea. They identify a specific point mutation in the cfr ORFs (C to A) that results in the Q148K mutation. They provide evidence that this mutation is responsible for the linezolid-susceptible phenotype in these isolates. The well-designed and well-executed study provide valuable insights into the molecular genetic mechanism of linezolid resistance in staphylococci.

However, a few critical points are relevant in response to the authors' findings. Firstly, the study is limited because it only examines a small number of isolates from pig farms in Korea, and the results may not be generalizable to other staphylococcal populations. Additionally, the study does not address the potential for the emergence of different resistance mechanisms in these isolates, as the authors only focus on the Q148K mutation in cfr. Furthermore, the study does not consider the potential impact of the identified mutation on the fitness of the bacteria or the potential for the mutation to spread to other bacteria.

Overall, the study provides important insights into the molecular genetic mechanism of linezolid resistance in staphylococci, and the authors' conclusions are well-supported by their data. However, the authors must conduct further research to fully understand the potential impact of the Q148K mutation on staphylococcal populations and the potential for other resistance mechanisms.

Author Response

The authors thoroughly examine the mechanism underlying linezolid susceptibility in cfr-positive Staphylococcus aureus and non-aureus staphylococci (NAS) isolates from pig farms in Korea. They identify a specific point mutation in the cfr ORFs (C to A) that results in the Q148K mutation. They provide evidence that this mutation is responsible for the linezolid-susceptible phenotype in these isolates. The well-designed and well-executed study provide valuable insights into the molecular genetic mechanism of linezolid resistance in staphylococci.

Response: We appreciate this reviewer's favorable comment on our manuscript.

However, a few critical points are relevant in response to the authors' findings. Firstly, the study is limited because it only examines a small number of isolates from pig farms in Korea, and the results may not be generalizable to other staphylococcal populations. Additionally, the study does not address the potential for the emergence of different resistance mechanisms in these isolates, as the authors only focus on the Q148K mutation in cfr. Furthermore, the study does not consider the potential impact of the identified mutation on the fitness of the bacteria or the potential for the mutation to spread to other bacteria.

Response: We agree that a rather small number of isolates has been included in this study. In addition, since we focused on linezolid resistance in staphylococci in this study, other mechanisms of antimicrobial resistance and the potential for horizontal transmission of the mutation to other bacteria were not pursued in detail. These limitations have been addressed in the revised manuscript.

The impact of Q148K mutation on the bacterial fitness was evaluated, and no significant difference has been found. These data have been added in the revised manuscript (Figure S2).

Overall, the study provides important insights into the molecular genetic mechanism of linezolid resistance in staphylococci, and the authors' conclusions are well-supported by their data. However, the authors must conduct further research to fully understand the potential impact of the Q148K mutation on staphylococcal populations and the potential for other resistance mechanisms.

Response: These limitations have been addressed in the revised manuscript.

Reviewer 2 Report

The article on title “Occurrence of cfr-positive linezolid-susceptible Staphylococcus aureus and non-aureus staphylococcal isolates from pig farms”,    lays out a very interesting finding to describe the effect of the cfr mutation (Q148K) on the PhLOPSA phenotype, which was determined using plasmid constructs expressing either the mutated (cfrQ148K) or nonmutated cfr genes, also genome and plasmids sequencing was used. The manuscript is well performed and easy to follow. I think that this research work adds important results to consider when cfr gene could be detected in bacterial strain from clinical origin. I recommend for acceptance.

1.-Abstract is adequately described.

2.- The introduction provide sufficient background and include relevant references.

3..- The methodology is adequately described.

4.- The results are clearly presented.

5.- The discussion and conclusions are supported by the results.

Author Response

Response: We thank this reviewer for his/her favorable comments on our manuscript.

Reviewer 3 Report

The manuscript is well written and well designed. However, there some points need to be addressed to improve the manuscript.

Line 84. The title of the chapter does not represent the result. The author wrote that there is no difference between resistant and susceptible strain in the promoter region. However, the title sounds like there are alterations that make the strain susceptible. It would be better to reformulate the title for this part.

Line 84-93. In this chapter of the results many data are not shown. Although no different or mutation, it is important to present the data obtained from the analyses. I suggest to put all the data for this chapter in the supplementary.

Table 1. In htis table, there so many acronyms used by the authors with no detail explanation provided. I suggest the authors provide the explanation of every acronyms used in the table.

Author Response

The manuscript is well written and well designed. However, there some points need to be addressed to improve the manuscript.

Response: We appreciate this reviewer for careful review of our manuscript.

Line 84. The title of the chapter does not represent the result. The author wrote that there is no difference between resistant and susceptible strain in the promoter region. However, the title sounds like there are alterations that make the strain susceptible. It would be better to reformulate the title for this part.

Response: The tile has been reformulated as suggested in the revised manuscript.

Line 84-93. In this chapter of the results many data are not shown. Although no different or mutation, it is important to present the data obtained from the analyses. I suggest to put all the data for this chapter in the supplementary.

Response: Those results have been added to the supplementary data as Figure S1 and Table S2 in the revised manuscript.

Table 1. In this table, there so many acronyms used by the authors with no detail explanation provided. I suggest the authors provide the explanation of every acronyms used in the table.

Response: Explanations for the acronyms in Table 1 have been added in the revised manuscript as suggested.

Reviewer 4 Report

In this article Gi Yong Lee et al. provide the important findings such as cfr-linezolid susceptible S.aureus and non-aureus staphylococcal isolates from pig forms and identified point mutations responsible for phenotypes susceptible to PhLOPSAantimicrobial agents . 

The manuscript was well written, the experiments well designed, and the conclusions are appropriate. I believe that the findings of the manuscript are of sufficient novelty and breadth to merit publication in antibiotics Journal. 

I have no major concerns with this paper in its current form.

Author Response

Response: We appreciate this reviewer for his/her favorable comments on our manuscript.

Round 2

Reviewer 3 Report

The manuscript is now much more improved and publishable.